# User intent driven retrieval augmented generation frameworks for auto-assisting compliance questionnaires

**First Author**[1] , **Second Author**[1] , **Third Author**[1]

[1]First Affiliation

{first, second, third}@example.com

## Abstract

AI models in production can pose risks related to ethics, regulations and compliance. Compliance frameworks and policies in organisations are fundamental in managing these risks. Questionnaires are an important tool adopted by organisations where owners or users of these models provide predefined information for review prior to deploying/using these AI models which can be mechanical and time-consuming. This paper discusses a retrieval augmented generation (RAG) framework to assist the end-user fill these questionnaires. In particular, early results show that one-shot human-in-the-loop RAG provides significant performance improvement in auto-assisting as compared to a traditional RAG model or a direct LLM model.

## 1 Introduction

The demand for deploying AI models on the cloud is steadily increasing across various domains. The advent of Large Language Models (LLMs) has significantly expanded the range of applications for AI. While these AI systems offer substantial societal benefits, ensuring their responsible deployment and monitoring for drifts is crucial to address concerns related to transparency, bias, compliance, and ethical implications. A range of tools and frameworks have been developed to assess the real-world impact of AI systems (ex. [Zhang *et al.*, 2023]).

As organisations adopt responsible AI frameworks into practice, questionnaires play a key role in the compliance process. They serve to ensure AI systems adhere to the company's internal governance guidelines, can be used to identify risks or concerns in AI system approval and contribute to stakeholder awareness through transparency. The questionnaires typically involve end-users providing responses that help evaluate the risks of an ML or LLM model that can be deployed for a given use case ([Raji *et al.*, 2020]). However, filling these questionnaires can be mundane and time-consuming for the end-user. This paper presents novel RAG based approaches to auto-fill or assist the end-user to fill compliance questionnaires based on user intent.

The rest of the paper is as follows. Section 2 provides a short literature survey of various methods and applications to auto-filling questionnaires followed by research overview in intent based AI. The architecture of the RAG platform for compliance questionnaires is provided in Section 3. Then, an evaluation of the framework is studied in Sections 4-6 by describing the datasets, evaluation metrics and results of the experiments respectively. Finally, the conclusion and future work is provided in Section 7.

## 2 Literature review

Numerous attempts have been made to auto-fill questionnaires in various fields like medicine, personality assessment, academic institutions etc ([Toudeshki *et al.*, 2022; Srivastava *et al.*, 2012; Puspitasari *et al.*, 2018] and citations within). The focus of these papers is to apply natural language inference (NLI) on text corpus or dialog contexts. For example, answering questionnaires require multiple interactions of the chat-bots with the end-user in [Toudeshki *et al.*, 2022] and extensive material from social media posts in [Spartalis *et al.*, 2021].

The emergence of LLMs has sparked significant advancements in NLI ([Chang *et al.*, 2023]). RAG frameworks, as particularly shown in [Wu *et al.*, 2024], have been successfully employed to fix hallucinations and provide up-to-date information to improve model accuracy. However, [Wu *et al.*, 2024] have shown that RAG documents are not strictly adhered to by LLM models if it significantly deviates from the prior knowledge of LLM models. Hence, it is important to understand the model behaviour in the presence of RAG document sources as well to ensure that the model inferences are trustworthy and accurate.

On the other hand, intent based networking is a novel concept that has been employed to configure, manage, and monitor networks based on user-intent which is usually provided as a couple of short sentences or phrases ([Leivadeas and Falkner, 2022] and references within). This concept has mainly been limited to networking community.

In this paper, in contrast to the works listed here, we present a novel methodology to use intents as an input for RAG based LLMs to auto-assist the end-user to fill out compliance questionnaires.

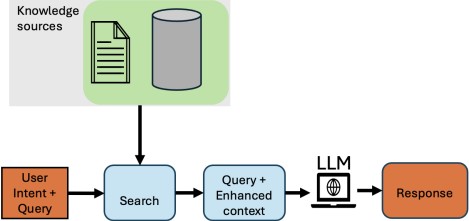

Figure 1: RAG based approach for auto-assisting compliance questionnaires.

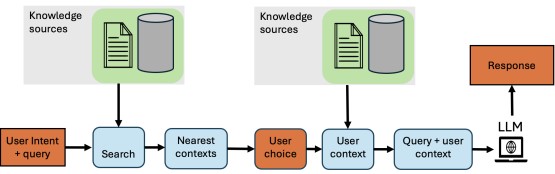

Figure 2: HITL RAG based approach for auto-assisting compliance questionnaires.

## 3 Architecture

The auto-assist framework expects user-intent that broadly defines the purpose of using LLMs or AI models as input. Further, a database of use-cases are stored as knowledge sources using historical records of user-intents and also synthetically generated using generative AI. We consider three main approaches to answering the questionnaires based on user-intent:

- Direct LLM (zero-shot): User intents are provided directly to LLMs and queries are answered directly based on the query and user-intent.

- RAG based LLM (single-shot): Vector database or knowledge sources of synthetic data is searched to find the closest match to the user-intent. The closest matching document is then used along with the query to auto-assist the answers for the questionnaire (Fig. 1).

- User driven RAG based LLM (human-in-the-loop (HITL) single shot): In this case, vector database or knowledge sources of synthetic data is searched to find the top k closest matches to the user-intent. The closest matching document is chosen by the end-user. This document is then used along with the query to auto-assist the answers for the questionnaire (Fig. 2).

## 4 Dataset

The dataset consists of 600 real intents provided by end users. Each user was asked to write a use case on how LLM can help them in their work. The users come from diverse professional backgrounds, resulting in intents that expand across domains, such as Customer service/support, Technical, Code/software engineering, Sales, Information retrieval, Strategy and others. The example user intents are shown in Table 1. We asked a

Table 1: User Intents from the dataset. Each example represents a particular domain type. Example (a) is from Strategy, example (b) is from Technical, and example (c) is from Customer service/support.

| User Intents |
|---|
| a. Generate optimized workflows, resource allocation plans, and process improvements for back-office operations |
| b. generate a list of SOE optimized keywords for a database platform |
| c. Support non-native customers in their mother tongue by providing content in a different language. |

Table 2: Example Synthetic Intents generated for the RAG based approach.

| Synthetic Intents |
|---|
| LLMs can be integrated into customer support chatbots to provide instant, accurate, and personalized responses to customer inquiries, helping to resolve issues quickly and efficiently. |
| LLMs can be trained on domain-specific text data to generate tailored medical reports, helping healthcare professionals save time and improve the accuracy of diagnoses. |
| LLMs can optimize supply chain management by analyzing demand, production, and logistics data to identify bottlenecks and suggest improvements. |

separate group of users to annotate these intents for each compliance question as shown in Table 3. In this paper, for every question type, we considered four domains from the dataset - Customer service/support, Technical, Information retrieval, and Strategy, totalling 81 use cases.

We performed experiments mainly with four main categories of questions: Dropdown - given a question and list of options, select an appropriate option based on the intent; Binary - based on the intent, state whether the given question evaluates to yes or no (true/false); Freeform - a descriptive answer is needed. We also compared the descriptive answers of Freeform questions via a fourth category - Compare, where we asked a different LLM to compare the response given by an LLM with the ground truth. Table 3 display each question type and the prompts.

## 5 Evaluation

The experiments used the LM Evaluation Harness [Gao *et al.*, 2023] to evaluate large language model responses. LM Evaluation tool can provide a range of metrics and benchmarks that comprehensively assess a wide range of LLM capabilities. It is entirely template-based and uses Unitxt [Bandel *et al.*, 2024], an open-source Python library that provides a consistent interface and methodology for defining datasets, preprocessing, and the metrics used to evaluate the results. For the experiments mentioned in this paper, we created four Unitxt templates, one each for the four question types. All three approaches, Direct LLM, RAG Basd LLM, and User-driven RAG-based LLM, use Unitxt templates to prepare the input questions, tune prompts, and pre and post-process the inputs and outputs. We ran these templates through LM Evaluation tool to obtain rouge metrics and accuracy.

## 6 Results

To test the model architectures, we generate synthetic data in four domains, namely, information retrieval, technical, customer service and strategy. We generate 10 synthetic use-cases for each domain and corresponding details on the

Table 3: Question types with prompts. Every prompt is formatted with the intent and question before being submitted to the LLM.

| Question type | Prompt |
|---|---|
| Dropdown | Based on the context given below, choose carefully the best option from the given options in the question. Give ouptut only from the options given.

context:
LLMs can be integrated into customer support chatbots to provide instant, accurate, and personalized responses to customer inquiries, helping to resolve issues quickly and efficiently. This is a Customer service/support use request.

question:
What domain does your use request fall under? Customer service/support, Technical, Information retrieval, Strategy, Other |
| Binary | Read the context given below and then answer the question in a single word. Answer Yes or No.

context:
LLMs can be integrated into customer support chatbots to provide instant, accurate, and personalized responses to customer inquiries, helping to resolve issues quickly and efficiently.

question:
Does the context include personal information |
| Freeform | Read the context given below carefully and then answer the question briefly.

context:
LLMs can be integrated into customer support chatbots to provide instant, accurate, and personalized responses to customer inquiries, helping to resolve issues quickly and efficiently.

question:
Please detail the input to be sent to the model. |
| Compare | context 1:The LLM will analyze news articles, identifying key topics, trends, and sources. It will then provide summaries, analysis, or insights, helping journalists and media professionals stay informed and make better decisions.

context 2:The input to the model would typically include a text or a set of text documents, such as news articles, as the primary source of information. The model would then analyze and process this text to identify key topics, trends, and sources.

State whether context 1 and context 2 given above are: Same, Similar, Different |

Table 4: Rouge Metric for LLMs evaluated using the question type Dropdown and approach types described in Section 3

| LLM | Approach | rouge1 | rouge2 | rougeL | rougeLsum |
|---|---|---|---|---|---|
| google/flan-ul2 | Direct LLM | 0.40 | 0.16 | 0.40 | 0.40 |
| | RAG based LLM | 0.49 | 0.28 | 0.49 | 0.49 |
| | User driven RAG based LLM | 0.78 | 0.44 | 0.78 | 0.78 |
| google/flan-t5-xl | Direct LLM | 0.61 | 0.32 | 0.61 | 0.61 |
| | RAG based LLM | 0.48 | 0.17 | 0.48 | 0.48 |
| | User driven RAG based LLM | 0.69 | 0.23 | 0.69 | 0.69 |
| meta-llama/llama-2-70b | Direct LLM | 0.39 | 0.34 | 0.39 | 0.39 |
| | RAG based LLM | 0.59 | 0.28 | 0.59 | 0.59 |
| | User driven RAG based LLM | 0.91 | 0.44 | 0.91 | 0.91 |
| ibm/granite-13b-lab-incubation | Direct LLM | 0.43 | 0.25 | 0.43 | 0.43 |
| | RAG based LLM | 0.59 | 0.28 | 0.59 | 0.59 |
| | User driven RAG based LLM | 0.93 | 0.44 | 0.93 | 0.93 |

Table 5: Rouge Metric for LLMs evaluated using the question type Binary (Yes/No) and approach types described in Section 3

| LLM | Approach | rouge1 | rouge2 | rougeL | rougeLsum |
|---|---|---|---|---|---|
| google/flan-ul2 | Direct LLM | 0.65 | 0.0 | 0.65 | 0.65 |
| | RAG based LLM | 0.64 | 0.0 | 0.64 | 0.64 |
| | User driven RAG based LLM | 0.69 | 0.0 | 0.69 | 0.69 |
| google/flan-t5-xl | Direct LLM | 0.62 | 0.0 | 0.62 | 0.62 |
| | RAG based LLM | 0.67 | 0.0 | 0.67 | 0.67 |
| | User driven RAG based LLM | 0.69 | 0.0 | 0.69 | 0.69 |
| meta-llama/llama-2-70b | Direct LLM | 0.54 | 0.0 | 0.54 | 0.54 |
| | RAG based LLM | 0.86 | 0.0 | 0.86 | 0.86 |
| | User driven RAG based LLM | 0.88 | 0.0 | 0.88 | 0.88 |
| ibm/granite-13b-lab-incubation | Direct LLM | 0.64 | 0.0 | 0.64 | 0.64 |
| | RAG based LLM | 0.77 | 0.0 | 0.77 | 0.77 |
| | User driven RAG based LLM | 0.86 | 0.0 | 0.86 | 0.86 |

inputs, privacy aspects etc for each of the use-cases respectively. Example synthetic intents are shown in Table 2. We checked the performance of filling the questionnaire using the three approaches described in Section 3 using the user intents as inputs and the synthetic use-cases as knowledge sources for RAG based LLM. We present the evaluation metrics using four LLMs: google/flan-ul2, google/flan-t5-xl, meta-llama/llama-2-70b, and ibm/granite-13b-lab-incubation. The rouge scores were recorded for the Binary, Dropdown, Freeform questions, and accuracy scores for the Compare question. We assess these scores across the three approaches described in Section 3.

As the original intents are fragmented, task-centric and particular to a user's domain type, no single LLM or prompt can directly predict the answers to compliance questionnaires, with acceptable accuracy, based on the intents alone. Our RAG-based approaches can better understand these intents and further provide improvisation to the predictions using human input. The results highlight a substantial advantage of using the user-driven RAG-based LLM approach over the RAG-based LLM approach and, subsequently, the RAG-based LLM approach over the Direct LLM approach. Tables 4 and 5 show rouge scores of four LLMs benchmarked on the Dropdown and Binary question, respectively. The example queries for Dropdown and Binary questions are shown in Table 3.

RAG-based LLMs offer a definitive edge over Direct LLMs. RAG's knowledge base, which spans multiple con-

texts, helps them incorporate prior knowledge, fill in missing pieces, and extrapolate knowledge beyond the query. However, this advantage vanishes quickly when conflicting information is present across RAG documents. User-based RAG LLMs can quickly mitigate this issue by bringing in the user's inherent knowledge about the context. In Tables 4 and 5, the user-driven RAG-based rouge score for all four LLMs is dominant by a significant margin over the other two. This result establishes that the user-driven RAG-based LLM approach is highly effective at predicting responses to questions, aided by the synthetic context and human involvement, with predictions moderately consistent with the end user's original intent.

Table 6 shows rouge scores for a Freeform question. The example query is given in Table 3. The rouge metric is based on individual word overlap and ordering consecutive words and phrases. Freeform questions have descriptive answers to the input query. The LLM models, in this case, can present the same output as the ground truth but may have different wordings and/or context, making the rouge extremely low. Hence, we report the accuracy score of the Freeform question using the LLM comparison question type. Using a different LLM, we compare LLM output and the ground truth to determine whether they are *Same*, *Similar* or *Different*. We swap the order of LLM output and the ground truth to remove the comparison bias and present the scores in Table 7.

Table 6: Rouge Metric for LLMs evaluated using the question type Freeform and approach types described in Section 3

| LLM | Approach | rouge1 | rouge2 | rougeL | rougeLsum |
|---|---|---|---|---|---|
| google/flan-ul2 | Direct LLM | 0.04 | 0.01 | 0.04 | 0.04 |
| | RAG based LLM | 0.12 | 0.04 | 0.12 | 0.12 |
| | User driven RAG based LLM | 0.18 | 0.07 | 0.17 | 0.17 |
| google/flan-t5-xl | Direct LLM | 0.03 | 0.0 | 0.03 | 0.04 |
| | RAG based LLM | 0.11 | 0.03 | 0.11 | 0.11 |
| | User driven RAG based LLM | 0.15 | 0.06 | 0.15 | 0.15 |
| meta-llama/llama-2-70b | Direct LLM | 0.12 | 0.03 | 0.10 | 0.10 |
| | RAG based LLM | 0.20 | 0.06 | 0.16 | 0.15 |
| | User driven RAG based LLM | 0.21 | 0.07 | 0.18 | 0.18 |
| ibm/granite-13b-lab-incubation | Direct LLM | 0.11 | 0.01 | 0.09 | 0.09 |
| | RAG based LLM | 0.14 | 0.03 | 0.11 | 0.11 |
| | User driven RAG based LLM | 0.16 | 0.04 | 0.13 | 0.13 |

Table 7: Accuracy Metric for LLMs evaluated using the question type Compare and approach types described in Section 3

| LLM | Approach | prediction and ground truth | ground truth and prediction |
|---|---|---|---|
| google/flan-ul2 | Direct LLM | 0.15 | 0.12 |
| | RAG based LLM | 0.57 | 0.43 |
| | User driven RAG based LLM | 0.80 | 0.70 |
| google/flan-t5-xl | Direct LLM | 0.29 | 0.23 |
| | RAG based LLM | 0.40 | 0.40 |
| | User driven RAG based LLM | 0.58 | 0.55 |
| meta-llama/llama-2-70b | Direct LLM | 0.30 | 0.32 |
| | RAG based LLM | 0.60 | 0.63 |
| | User driven RAG based LLM | 0.81 | 0.78 |
| ibm/granite-13b-lab-incubation | Direct LLM | 0.20 | 0.25 |
| | RAG based LLM | 0.30 | 0.33 |
| | User driven RAG based LLM | 0.55 | 0.55 |

User-driven RAG-based LLM again tops the accuracy scores regardless of the context order.

It is also important to note that the tables are not indicative of one LLM outperforming the others since we did not perform any prompt engineering to improve the performance of individual LLMs. We used the same prompt for all the LLMs since the focus of this study is to compare the different frameworks for answering the questionnaire rather than improve the performance of individual LLMs.

# 7 Conclusion

We presented a novel intent based RAG framework to auto-assist the end-user to complete compliance questionnaires and minimize the effort to responsibly deploy AI models into production. We noticed that the HITL RAG based approach provides the best performance in auto-assisting. Future work will concentrate on improved query rewriting for user intents to further improve the performance.

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
