# OpenReview forum: "User intent driven retrieval augmented generation frameworks for auto-assisting compliance questionnaires"
_ijcai.org/IJCAI/2024/Workshop/TIDMwFM — IJCAI TIDMwFM 2024 Poster_

### Official Review · Reviewer_JLKF · 2024-06-19

**Rating:** 7
**Confidence:** 4

**Review:**

This paper explores an innovative approach to streamline the process of filling compliance questionnaires using Retrieval Augmented Generation (RAG) frameworks. This research is particularly relevant to the theme of "Trustworthy Interactive Decision-Making with Foundation Models Workshop," as it seeks to enhance the reliability and efficiency of AI-assisted compliance procedures.

The authors present a comprehensive comparison of three methodologies: Direct LLM (zero-shot), RAG based LLM (single-shot), and User driven RAG based LLM (HITL single-shot). The HITL approach, which incorporates user feedback to refine the model's outputs, shows a significant performance improvement over the traditional RAG and direct LLM methods. This is particularly evident in the rouge and accuracy metrics across various question types, demonstrating the framework's effectiveness in understanding and responding to user intents accurately.
The dataset, comprising 600 real user intents across diverse professional domains, adds robustness to the study. The use of synthetic data to generate additional intents further supports the model's ability to generalize across different contexts. The results indicate that user-driven RAG-based models can better handle the complexities and nuances of compliance questionnaires, providing outputs that align closely with user expectations and regulatory requirements.

One of the key contributions of this paper is its novel approach to combining user intents with RAG frameworks. This methodology not only improves the auto-assisting capabilities of compliance questionnaires but also ensures that the outputs are contextually relevant and trustworthy. The incorporation of human-in-the-loop mechanisms highlights the potential for enhancing model accuracy through user feedback, a critical aspect for maintaining compliance and ethical standards in AI deployments.

---

### Official Review · Reviewer_rq2a · 2024-06-21

**Rating:** 7
**Confidence:** 3

**Review:**

This paper presents a novel approach to auto-assist users in filling out compliance questionnaires for AI model deployment using retrieval augmented generation (RAG) frameworks based on user intent. The authors compare three approaches: direct LLM, RAG-based LLM, and user-driven RAG-based LLM (human-in-the-loop).

Strengths:

1. The paper addresses an important practical problem in AI compliance and governance.

2. The proposed user-driven RAG approach shows promising results compared to baseline methods.

3. Clear explanation of the methodology.

Limitations/Points for Improvement:

1. The sample size (81 use cases) for the main experiments seems relatively small.

2. Limited discussion on the potential biases or limitations of the synthetic data generation process.

3. The paper could benefit from more discussion on the practical implications and potential deployment challenges.

4. More details on the prompt engineering process could be valuable.

---

### Decision · Program_Chairs · 2024-06-24

Accept (Poster)